# AutoGluon-Multimodal (AutoMM): Supercharging Multimodal AutoML with Foundation Models

Zhiqiang Tang[1,*]  Haoyang Fang[1]  Su Zhou[1]  Taojiannan Yang[1]  Zihan Zhong[1]  Tony Hu[1]
Katrin Kirchhoff[1]  George Karypis[1]

[1]Amazon Web Services
* Correspondence to: Zhiqiang Tang <zqtang@amazon.com>

**Abstract**  AutoGluon-Multimodal (AutoMM) is introduced as an open-source AutoML library designed specifically for multimodal learning.[1] Distinguished by its exceptional ease of use, AutoMM enables fine-tuning of foundation models with just three lines of code. Supporting various modalities including image, text, and tabular data, both independently and in combination, the library offers a comprehensive suite of functionalities spanning classification, regression, object detection, semantic matching, and image segmentation. Experiments across diverse datasets and tasks showcases AutoMM's superior performance in basic classification and regression tasks compared to existing AutoML tools, while also demonstrating competitive results in advanced tasks, aligning with specialized toolboxes designed for such purposes.

## 1 Introduction

Automated machine learning (AutoML) [Yao et al., 2018, Zöller and Huber, 2021, He et al., 2021b] promises to streamline the process of translating raw data into accurate predictions, minimizing the need for extensive human intervention and expertise, though a noticeable gap still exists in some domains [Schmarje et al., 2021, Parisi et al., 2019, Chan et al., 2020]. By encapsulating best practices in machine learning—from data preprocessing [Gada et al., 2021] to model selection [Arango et al., 2023a], training [Falcon, 2019], and deployment [Paleyes et al., 2022]—AutoML frameworks aim to democratize machine learning capabilities, enabling both technical and non-technical users to develop high-performing models efficiently. This scalability of expertise empowers practitioners to tackle a wide array of tasks without requiring deep knowledge of machine learning techniques.

The continual evolution of machine learning techniques, particularly the advent of foundation models [Bommasani et al., 2021] that are pre-trained on large-scale datasets and are applicable to a wide array of downstream tasks have revolutionized fields such as computer vision [Dosovitskiy et al., 2020, Liu et al., 2021] and natural language processing [Devlin et al., 2018, Liu et al., 2019]. Fine-tuning these models for specific domains is crucial for extending their utility to end-users, yet dedicated open-source AutoML toolboxes for this purpose remain scarce. In addition, existing well-known open-source AutoML toolboxes predominantly focus on basic classification and regression tasks with tabular data [Thornton et al., 2013, Feurer et al., 2015, Olson and Moore, 2016, Erickson et al., 2020, LeDell and Poirier, 2020, Zimmer et al., 2021], overlooking the complexities of real-world problems that often entail multiple modalities [Baltrušaitis et al., 2018]. For instance, tasks like webpage analysis typically involve processing image, text, and tabular data concurrently, with objectives ranging from object detection [Zou et al., 2023] to semantic matching [Reimers and Gurevych, 2019]. Despite the availability of specialized tools for individual tasks, there is a notable gap in the AutoML community regarding unified frameworks capable of handling diverse modalities and tasks seamlessly.

---

[1]https://github.com/autogluon/autogluon

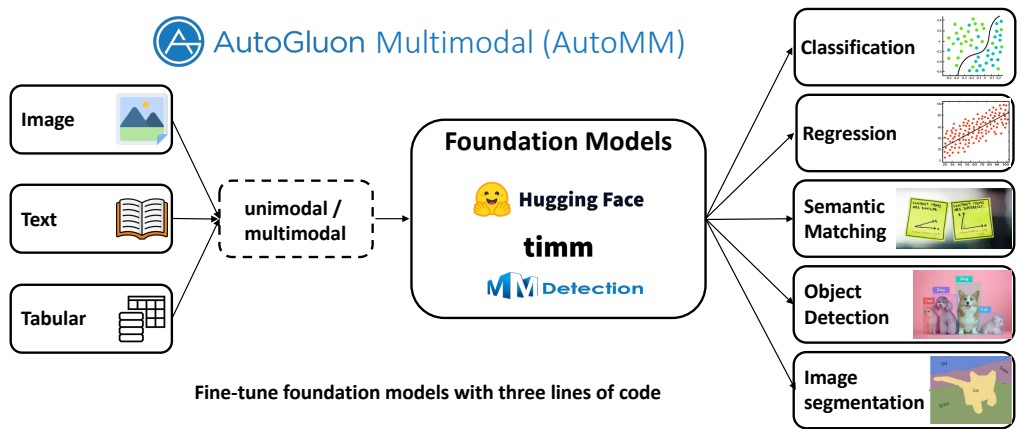

Figure 1: AutoMM introduction: Supporting both unimodal and multimodal data (**Left**), AutoMM enables seamless fine-tuning of foundation models (**Middle**) for basic classification/regression as well as advanced tasks (**Right**).

To address these dual challenges, we introduce AutoGluon-Multimodal (AutoMM)—a Python-based open-source AutoML framework tailored for multimodal learning with foundation models. Embedded within the AutoGluon ecosystem [Erickson et al., 2020, Shchur et al., 2023], AutoMM enables users to fine-tune foundation models effortlessly on domain-specific data with just three lines of code. Leveraging popular model repositories such as Huggingface/transformers [Wolf et al., 2020], TIMM [Wightman, 2019], and MMDetection [Chen et al., 2019], AutoMM supports a wide range of modalities including text, image, and tabular data, facilitating tasks such as classification, regression, object detection [Zou et al., 2023], semantic matching [Reimers and Gurevych, 2019], and image segmentation [Minaee et al., 2021]. Figure 1 outlines the AutoMM framework, illustrating its key functionalities.

The evaluation of AutoMM presents two primary challenges: the absence of established benchmark datasets covering multiple modalities and tasks, and the scarcity of competing AutoML libraries with comparable functionalities. To address the former, we curated a benchmark comprising 55 publicly available datasets spanning diverse modalities and tasks, prioritizing real-world applications over academic datasets for a more robust evaluation of AutoML toolboxes. Mitigating the latter challenge, we conducted a comprehensive evaluation encompassing basic and advanced tasks. In basic classification and regression tasks across 24 unimodal and multimodal datasets, AutoMM outperformed AutoKeras [Jin et al., 2023] significantly. For advanced tasks of semantic matching and semantic segmentation, comparisons with task-specific open-source libraries demonstrated comparable performance. These findings underscore the potential of AutoMM as a comprehensive solution for practitioners seeking automated solutions across various modalities and tasks.

## 2 Related Work

AutoML [Yao et al., 2018, Zöller and Huber, 2021, He et al., 2021b] has emerged as a pivotal area of research and development, aiming to democratize machine learning by automating the complex processes of model selection [Arango et al., 2023a], hyperparameter tuning [Bischl et al., 2023], and feature engineering [Zheng and Casari, 2018]. Traditional AutoML methods primarily focus on basic classification and regression tasks for tabular data. For example, Auto-Sklearn [Feurer et al., 2015] is built on the scikit-learn library, and leverages bayesian optimization, meta-learning and ensemble construction to provide a robust solution for classification and regression tasks. TPOT [Olson and Moore, 2016] also uses scikit-learn as its ML library and employs genetic algorithms to optimize entire ML pipelines, including preprocessing and modeling steps. H2O AutoML [LeDell

and Poirier, 2020] provides an end-to-end platform for automating the machine learning process, including automatic training and ensemble of a diverse set of algorithms such as GBMs, Rnadom Forests, Deep Neural Network, GLMs. FLAML [Wang et al., 2021a] optimizes for low computational cost in the hyperparameter search.

Despite the significant strides made in traditional AutoML domains, the exploration of AutoML capabilities for handling multimodal inputs and tackling advanced tasks remains relatively nascent. While several conventional AutoML frameworks [Vakhrushev et al., 2021, Feurer et al., 2015] are equipped to process multimodal data, their methodologies predominantly hinge on traditional machine learning algorithms. AutoKeras, an AutoML framework leveraging deep learning [Jin et al., 2023], broadens its applicability to multimodal inputs. However, this functionality requires users to manually specify and preprocess for each modality, which increases the likelihood of errors. Additionally, AutoKeras falls short in addressing more complex downstream tasks, such as object detection [Zou et al., 2023] and image segmentation [Minaee et al., 2021], limiting its utility in advanced applications. LightAutoML [Vakhrushev et al., 2021] is a lightweight framework which provides fast hyperparameter search and focuses on limited models such as gradient boosted decision trees and linear models for multimodal data. However, its support for multimodal input is restricted to tabular-image and tabular-text only, which limits its usage in real-world applications.

There are also AutoML methods based on foundation models. For example, Quick-Tune [Arango et al., 2023b] proposes how and which models to fine-tune from a pre-trained zoo. TabPFN [Hollmann et al., 2022] proposes to use transformers to solve small tabular classification problems without hyperparameter tuning. OptFormer [Chen et al., 2022] leverages transformers to learn universal hyperparameter optimizers. However, these methods all focused on unimodal data. In contrast, AutoMM is designed to proficiently manage multimodal inputs, including image, text, and tabular, in a unified framework with three lines of codes. Beyond the basic classification and regression tasks, AutoMM also supports real-world application tasks such as object detection, image segmentation, semantic matching [Reimers and Gurevych, 2019], etc. At its core, AutoMM leverages the transformative power of recent foundation models [Bommasani et al., 2021], capitalizing on their exceptional transfer learning capabilities to achieve state-of-the-art results across a diverse array of tasks.

## 3 AutoMM

Supporting diverse data modalities and task types, while staying true to the AutoML philosophy, presents significant challenges. The primary obstacle lies in automating these processes through unified data format/processing, APIs, model design, and training workflow. In this section, we delineate the design and functionalities of AutoMM.

### 3.1 Data Format and Processing

AutoMM employs Pandas DataFrame [pandas development team, 2020] to consolidate various data modalities, including images, text, and tabular (numeric and categorical) data. DataFrames are ubiquitous in modern data analytics due to their flexibility and user-friendly nature. Essentially, a Pandas DataFrame organizes data into a 2-dimensional table, with rows representing individual samples and columns representing features. Each field within the DataFrame can accommodate different data types, such as numeric, categorical, text, image paths, bytearray images, or base64-encoded images. This versatility allows users to provide AutoMM with multimodal DataFrames containing any combination of modalities. Moreover, each modality can comprise multiple columns, facilitating complex data representations. As an AutoML toolbox, AutoMM adeptly handles raw and noisy data, alleviating users from the preprocessing burden.

Traditionally, addressing different modalities entails constructing independent data processing pipelines for each. However, such an approach leads to code redundancy and increased maintenance overhead. To mitigate this, we systematically analyzed these pipelines and abstracted

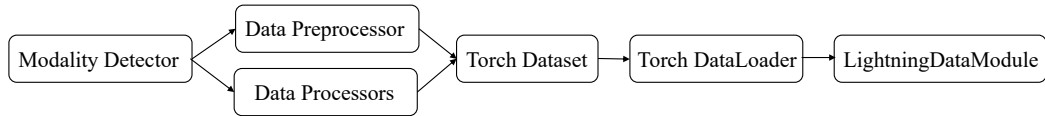

Figure 2: Interdependency among data modules: Each module from left to right serves as a prerequisite for the subsequent one.

their commonalities, resulting in a generalized pipeline. Our unified data pipeline comprises LightningDataModule [Falcon, 2019], torch [Paszke et al., 2019] DataLoader, torch Dataset, a modality detector, a data preprocessor, and arbitrary data processors. At a high level, the LightningDataModule invokes torch DataLoader to generate distinct data loaders for training, validation, and prediction stages. The creation of torch DataLoader necessitates a torch Dataset object, which, in turn, relies on the data preprocessor and data processors. These components are instantiated based on the detected modalities by the modality detector. The interdependence among these data modules is illustrated in Figure 2.

The data preprocessor handles model-agnostic data processing tasks, such as filtering non-informative features, converting data types, handling null values, and normalizing numeric data. In contrast, data processors cater to model-specific processing requirements. Since multiple models may operate concurrently, with each potentially accepting multiple modalities as inputs, we create a dedicated data processor for each modality of every model. The data preprocessor conducts DataFrame-level preprocessing during torch Dataset initialization, while data processors perform online transformation of individual samples during data loading. Additionally, we define data collator functions to aggregate processed samples into mini-batches for model consumption.

Data processors are responsible for per-sample processing, encompassing data augmentation [Shorten and Khoshgoftaar, 2019, Shorten et al., 2021] and multi-field processing. Numeric fields are concatenated into a single vector, while categorical fields are encoded separately using torch nn.Embedding. Image processing involves augmentation (e.g., TrivialAugment [Müller and Hutter, 2021]), tensor conversion, and normalization. In cases where multiple images per sample are present, AutoMM processes each image individually before stacking them. Text data undergoes tokenization, with each text field tokenized separately and concatenated into a unified token sequence. To address sequence length constraints imposed by models, we truncate the sequence by iteratively removing tokens from the longest text field until compliance is achieved.

### 3.2 APIs

AutoMM streamlines the fine-tuning of foundation models on unimodal or multimodal data through a concise API interface. To illustrate, let's focus on basic classification/regression tasks using a DataFrame stored in "train.csv" with the label column "label". With just three lines of code, users can import `MultiModalPredictor`, initialize an object, and begin training:

```
from autogluon.multimodal import MultiModalPredictor
predictor = MultiModalPredictor(label="label")
predictor.fit("train.csv")
```

During training, AutoMM automatically infers the problem type (e.g., binary classification, multiclass classification, or regression), partitions the data into training and validation sets, identifies data modalities, selects appropriate foundation models, and performs fine-tuning. The `fit()` method also offers additional customization options, such as controlling training time, customizing hyperparameters , or conducting hyperparameter optimization. Continuous training is supported, enabling sequential calls of `fit()` on incoming training data.

Upon completion of training, users can leverage various APIs for evaluation or inference:

```
score = predictor.evaluate("test.csv")
```

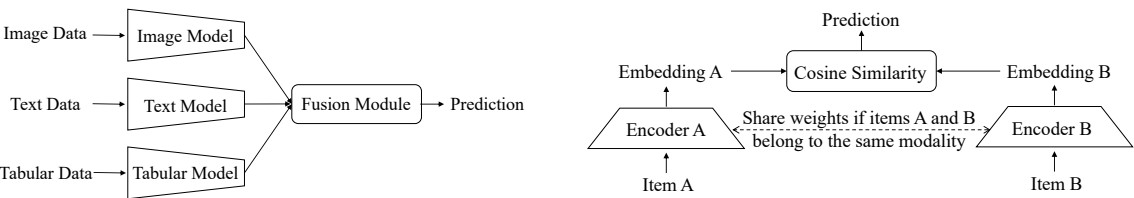

Figure 3: Late-fusion model.

Figure 4: Bi-encoder model.

```
2  predictions = predictor.predict("new.csv")
3  probabilities = predictor.predict_proba("new.csv")
4  embeddings = predictor.extract_embedding("new.csv")
```

These APIs facilitate model evaluation, prediction, probability estimation (for classification tasks), and feature embedding extraction. Notably, prediction-related APIs (`predict()`, `predict_proba()`, and `extract_embedding()`) do not necessitate labels, enhancing their utility for deployment purposes. Predictors can be saved and loaded for future use via the `save()` and `load()` methods:

```
1  predictor.save("save_path")
2  predictor = MultiModalPredictor.load("save_path")
```

The `load()` method also supports training resumption in case of training interruptions by specifying `resume=True` and providing the path to the interrupted predictor. Calling `fit()` afterwards can resume training from the last saved checkpoint. Furthermore, we provide various additional resources on the official website auto.gluon.ai, including installation instructions, hands-on tutorials, and a cheatsheet summarizing the main features.

### 3.3 Models

AutoMM aims to apply foundation models across diverse real-world scenarios. Foundation models undergo pretraining on extensive datasets before fine-tuning on smaller downstream labeled datasets. However, this fine-tuning necessitates alignment between downstream data modalities and those encountered during pretraining. For instance, while BERT [Devlin et al., 2018] can fine-tune on text-only tasks, it cannot handle text+tabular data directly. Despite the proliferation of foundation models, their pretraining modalities typically span image-only [Dosovitskiy et al., 2020], text-only [He et al., 2021a], or image+text domains [Radford et al., 2021]. Yet, practical applications frequently involve more combinations such as image+text+tabular. To bridge this gap, AutoMM adopts a late-fusion architecture, scalable to accommodate arbitrary modality combinations. The late-fusion framework, shown in Figure 3, integrates independent backbones for images, text, and tabular data, followed by a fusion module (e.g., MLP or transformer [Vaswani et al., 2017] layers) for feature fusion. In scenarios involving a single modality, the fusion module is bypassed. Notably, adding support for a new modality seamlessly enables integration with existing ones.

AutoMM offers extensive model zoo compatibility, including Huggingface/transformers [Wolf et al., 2020], TIMM [Wightman, 2019], and MMDetection [Chen et al., 2019]. These repositories contain lots of pretrained models with >25000, >700, and >300 respectively, as documented. Text models (e.g., Electra [Clark et al., 2020], Deberta [He et al., 2021a]) from Huggingface and image models (e.g., Swin Transformer [Liu et al., 2021], ViT [Dosovitskiy et al., 2020]) from TIMM are predominantly utilized. MMDetection supplies pretrained models for object detection tasks, such as YOLOX [Ge et al., 2021] and DINO [Zhang et al., 2022]. Pretrained models generally span a spectrum of sizes (e.g., ViT-large, ViT-base, ViT-small), each associated with distinct performance and resource considerations. AutoMM defines three preset levels (`best_quality`, `high_quality`, `medium_quality`) to accommodate varying performance-latency trade-offs. The selection of preset models is based on our internal benchmarking results, streamlining model selection for end-users.

For the basic classification/regression, we employ a two-stage selection process. Initially, we individually select the top 5 models for image, text, and tabular data based on their unimodal benchmarking performance. Subsequently, we conduct a random search of their combinations, along with other hyperparameters like learning rate and epochs, on multimodal benchmarks. The resulting best combination serves as the default models.

## 3.4 Training

By default, AutoMM trains only one late-fusion model with fixed hyperparameters (without optimization). These default hyperparameters, e.g., learning rate and weight decay, have been pre-determined through offline search across diverse benchmark datasets, assuming they are likely to yield reasonable performance on new (similar) datasets. While some AutoML toolboxes like Autosklearn 2.0 [Feurer et al., 2022] also use the pre-determined hyperparameters, our method differs in that it emphasizes offline tuning, rather than further hyperparameter optimization or model ensembling [Erickson et al., 2020]. Such methods often require training and evaluating numerous models, assuming each trial has relatively low cost. However, fine-tuning foundation models on multimodal data is generally resource-intensive and may not align with this assumption. To address practical concerns, we opt for offline hyperparameter search across our benchmark datasets, enabling end-users to experience efficiency and effectiveness. Detailed hyperparameters are provided in Section C for reference.

AutoMM leverages the Lightning framework [Falcon, 2019], built on top of PyTorch [Paszke et al., 2019], to streamline the training workflow. Lightning abstracts key training components into LightningDataModule, LightningModule, Trainer, and Callbacks, enhancing maintainability and scalability. LightningDataModule oversees data loading and processing, employing our data preprocessor and processors. LightningModule defines the model's forward pass and optimizer configuration. Encouraging modular design, our model is created outside LightningModule, being passed into it as needed. Trainer orchestrates the training process, offering extensive configuration options (e.g., gradient accumulation, training precision) and driving training through interaction with LightningDataModule and LightningModule objects. We also utilize Lightning's build-in Callbacks that can provide additional functionalities during the training loop, including logging, checkpointing, and early stopping.

Given the rapid expansion of foundation model sizes, fine-tuning these models poses challenges, particularly for users with limited computational resources. To mitigate this, AutoMM embraces parameter-efficient fine-tuning techniques (PEFT) [Houlsby et al., 2019]. These techniques, such as BitFit [Zaken et al., 2021] and LoRA [Hu et al., 2021], either optimize a tiny fraction of pretrained weights or introduce lightweight structures on top of fixed pretrained models to reduce memory footprint and training time while preserving performance. For instance, IA3 [Liu et al., 2022] facilitates fine-tuning of the Flan-T5-XL [Chung et al., 2022] model's encoder (1.2 billion parameters) on a single NVIDIA T4 GPU with 15 GB memory. Additionally, AutoMM supports distributed training (multi-GPU or multi-node) and low-precision (default 16-mixed) training [Micikevicius et al., 2017], further easing the burden posed by large models.

## 3.5 Deployment

Efficient deployment of trained models into production environments is crucial for real-world applications. To ensure seamless predictor loading in an offline environment, AutoMM pre-saves requisite artifacts—e.g., Huggingface model configuration—forestalling Internet access errors upon predictor invocation. Notably, in production settings, low inference latency, particularly for online inference, is paramount. Although our training hinges on Lightning modules adept at handling large sample sizes, these modules may impede inference when processing few samples. To address this, we furnish a realtime option, eschewing Lightning modules in favor of plain PyTorch models and data processing to expedite inference. Moreover, AutoMM integrates with NVIDIA TensorRT

[Vanholder, 2016], encompassing a deep learning inference optimizer and runtime renowned for low latency and high throughput. Additionally, accommodating varying image formats between training and deployment—e.g., image path in training versus image bytearray in deployment—AutoMM dynamically infers image sub-types during inference, ensuring seamless model deployment.

### 3.6 Advanced Tasks

AutoMM expands traditional AutoML beyond basic classification and regression tasks to encompass advanced functionalities. As of the time of writing, AutoMM supports tasks such as semantic matching, object detection, and semantic segmentation. Semantic matching involves assessing the similarity between two items, which can be two images, two texts, or an image-text pair. We utilize a bi-encoder design, illustrated in Figure 4, that independently encodes the two items in the embedding space before computing their semantic similarity. Object detection and semantic segmentation are both computer vision tasks. Object detection locates object instances within an image, while semantic segmentation categorizes each pixel in an image into a class or object. In terms of output, object detection provides object class labels and bounding boxes, whereas segmentation generates segmentation masks. Despite the complexity of these tasks, users can implement solutions with just three lines of code, albeit with the necessity to specify the `problem_type` argument when initializing the predictor object. AutoMM achieves these advanced functionalities by fine-tuning foundation models from Huggingface and MMDetection. For instance, it can employ CLIP [Radford et al., 2021] for image-text matching, DINO [Zhang et al., 2022] for object detection, and the Segment Anything Model [Kirillov et al., 2023] for semantic segmentation.

## 4 Experiments

In this experimental study, we aim to assess the efficacy of AutoMM compared to other leading AutoML solutions or task-specific toolboxes across a diverse range of tasks, including classification or regression, semantic matching, object detection, and semantic segmentation.

### 4.1 Classification and Regression

We start by experimenting how AutoMM performs on classification and regression tasks. We apply AutoMM to 24 real-life datasets with an assortment of tasks. The input in each task may involve any combinations of {Tabular, Image, Text}. The performance metrics we reported are *R-squared* ($R^2$), *F1_weighted* and *F1* for regression, multi-class and binary classification tasks respectively.

To the best of our knowledge, the only other AutoML framework that supports multimodal input is Auto-Keras, serving as the main baseline here. The results, which were run over 5 independent random seeds (22, 41, 54, 86, 92) shown in Table 1, indicate that AutoMM works out-of-the-box on all datasets with no prior knowledge or dataset-specific configuration. Compared to Auto-Keras, AutoMM performs better in all 24 datasets across different problem types and data modalities, with *statistical significance*. We also observe that the performance of Auto-Keras often vary drastically over different runs, while AutoMM provides substantially more consistency and reliability. Furthermore, AutoMM excels in ease of use: datasets can simply be loaded into a Pandas DataFrame, and modalities are automatically detected by AutoMM. For image modalities, AutoMM accepts both image paths and image bytearrays, without the need for further processing. In contrast, Auto-Keras requires that data be reformatted into numpy arrays by the user, and modalities must be explicitly specified, such as *autokeras.ImageInput()*, *autokeras.StructuredDataInput()*, and *autokeras.TextInput()*, which can introduce inconsistency and human errors. For further details on datasets, baseline methods, metrics, and additional setup information, please refer to the appendix.

### 4.2 Semantic Matching

For semantic matching, we compare with Sentence-Transformer [Reimers and Gurevych, 2019], which is a toolbox designed specifically for semantic matching tasks. We show the comparisons

| Dataset | Text | Image | Tabular | Problem Type | Metric | Auto-Keras | AutoMM |
|---|---|---|---|---|---|---|---|
| fashion_mnist | ✗ | ✓ | ✗ | Multiclass | F1_weighted↑ | 0.876(0.020) | **0.953**(0.002) |
| food101 | ✗ | ✓ | ✗ | Multiclass | F1_weighted↑ | 0.024(0.045) | **0.937**(0.001) |
| Stanford_cars | ✗ | ✓ | ✗ | Multiclass | F1_weighted↑ | 0.055(0.079) | **0.892**(0.002) |
| magnetic_tile_defects | ✗ | ✓ | ✗ | Multiclass | F1_weighted↑ | 0.627(0.171) | **0.956**(0.014) |
| European_flood_depth | ✗ | ✓ | ✗ | Binary | F1↑ | 0.750(0.017) | **0.790**(0.008) |
| Oxford_flowers | ✗ | ✓ | ✗ | Multiclass | F1_weighted↑ | 0.123(0.155) | **0.989**(0.003) |
| OxfordIIITPet | ✗ | ✓ | ✗ | Multiclass | F1_weighted↑ | 0.157(0.283) | **0.958**(0.003) |
| CD18_cellphone | ✗ | ✓ | ✗ | Regression | $R^2$ ↑ | -18.390(35.120) | **-1.843**(4.477) |
| HAM10000 | ✗ | ✓ | ✗ | Multiclass | F1_weighted↑ | 0.276(0.211) | **0.608**(0.014) |
| hateful_meme | ✓ | ✓ | ✗ | Binary | F1↑ | 0.572(0.099) | **0.596**(0.013) |
| petfinder | ✓ | ✓ | ✓ | Multiclass | F1_weighted↑ | 0.243 (0.040) | **0.408**(0.006) |
| memotion | ✓ | ✓ | ✗ | Multiclass | F1_weighted↑ | 0.297 (0.026) | **0.467**(0.013) |
| financial_news | ✓ | ✗ | ✗ | Multiclass | F1_weighted↑ | 0.678(0.027) | **0.874**(0.010) |
| MLDoc-11000 | ✓ | ✗ | ✗ | Multiclass | F1_weighted↑ | 0.916(0.006) | **0.978**(0.002) |
| gnad10 | ✓ | ✗ | ✗ | Multiclass | F1_weighted↑ | 0.521(0.029) | **0.899**(0.006) |
| MultiATIS-5000 | ✓ | ✗ | ✗ | Multiclass | F1_weighted↑ | 0.864(0.010) | **0.990**(0.003) |
| fb_dialog | ✓ | ✗ | ✗ | Multiclass | F1_weighted↑ | 0.982(0.003) | **0.992**(0.001) |
| SNIPS | ✓ | ✗ | ✗ | Multiclass | F1_weighted↑ | 0.049(0.018) | **0.990**(0.002) |
| ag_news | ✓ | ✗ | ✗ | Multiclass | F1_weighted↑ | 0.887(0.004) | **0.944**(0.001) |
| airbnb_melbourn | ✓ | ✗ | ✓ | Multiclass | F1_weighted↑ | 0.198(0.071) | **0.397**(0.011) |
| kick_start_funding | ✓ | ✗ | ✓ | Binary | F1↑ | 0.401 (0.151) | **0.609**(0.005) |
| cloth_review | ✓ | ✗ | ✓ | Regression | $R^2$ ↑ | 0.542(0.053) | **0.735**(0.004) |
| news_popularity | ✓ | ✗ | ✓ | Regression | $R^2$ ↑ | -1.306(1.863) | **0.014**(0.003) |
| California_house | ✓ | ✗ | ✓ | Regression | $R^2$ ↑ | -53757156.425 (55682587.109) | **0.944**(0.001) |

Table 1: Results for unimodal/multimodal classification and regression. The mean performance metrics and error bars (in parentheses) with 0.95 coverage are reported for both AutoMM and Auto-Keras. These numbers are estimated using 5 independent repeats with different random seeds using $1.96 \times std/\sqrt{\text{\# of repeats}}$. Boldface indicates results that are better than the competing framework with *statistical significance* at the level 0.05 using Wald test [Wald, 1943].

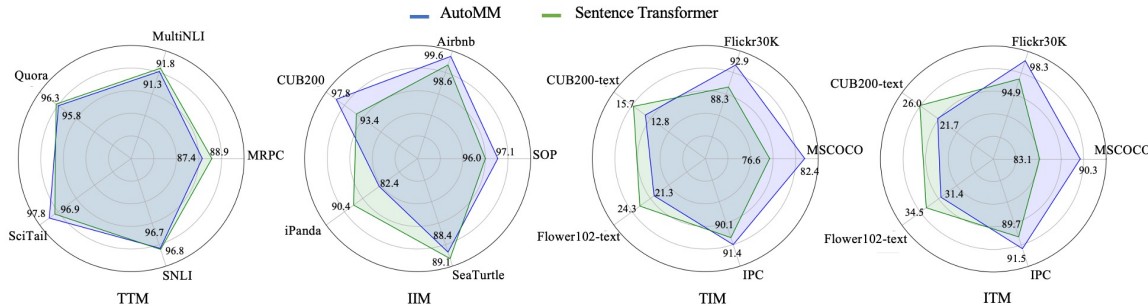

Figure 5: Comparison of AutoMM and Sentence-Transformer on Text to Text Matching (TTM), Image to Image Matching (IIM), Text to Image Matching (TIM), and Image to Text Matching (ITM).

on Text to Text Matching (TTM), Image to Image Matching (IIM), Text to Image Matching (TIM), and Image to Text Matching (ITM) tasks in Figure 5. AutoMM achieves competitive performance with Sentence Transformer, while being more user-friendly and straightforward to utilize. For Sentence Transformer, users have to manually select the optimal loss function for different tasks to achieve the best performance, which requires the user to have extensive domain expertise and a large amount of trial and error. However, AutoMM automates the entire pipeline while being competitive in different tasks. For further details on datasets, baseline methods, metrics, and raw results, please refer to the appendix.

| | | Vertex AI | | | Nvidia Tao | | | AutoMM | | |
|---|---|---|---|---|---|---|---|---|---|---|
| | | training time (hrs) | mAP | AP50 | training time (hrs) | mAP | AP50 | training time (hrs) | mAP | AP50 |
| Agriculture | plantdoc | 2.8 | / | 49.1 | 3.5 | 55.5 | 70.8 | **2.1(0.23)** | **58.7(0.38)** | **73.3(0.54)** |
| | deepfruits | / | / | / | **0.6** | 51.4 | 75.2 | **0.6(0.07)** | **71.3(0.56)** | **95.0(0.19)** |
| Medical | chest10 | 3.8 | / | 36.8 | 2.5 | **19.5** | **41.4** | **2.1(0.11)** | 18.3(0.28) | 37.2(1.2) |
| | deeplesion | / | / | / | 23.5 | 39.4 | 67.5 | **18.7(0.40)** | **40.4(0.24)** | **70.5(0.49)** |
| Domain Transfer | comic | 2.8 | / | 57.6 | 1.1 | 22.7 | 42.6 | **0.9(0.09)** | **42.3(0.72)** | **67.7(0.62)** |
| Remote Sencing | dota | 11.8 | / | 60.7 | 10.0 | 47.9 | 74.1 | **9.0(0.75)** | **51.3(0.14)** | **76.9(0.18)** |
| Autonomous Driving | kitti | / | / | / | 7.9 | **78.7** | **96.4** | **5.9(0.44)** | 71.5(0.76) | 95.4(0.14) |
| Infrared | thermal | / | / | / | **0.3** | 66.8 | 82.2 | **0.3(0.05)** | **82.9(0.88)** | **95.3(1.08)** |

Table 2: Comparison of AutoMM and baseline methods on object detection tasks in various domains. The mean performance metrics and error bars (in parentheses) with 0.95 coverage are reported for AutoMM. These numbers are estimated using 3 independent repeats with different random seeds using $1.96 \times \text{std}/\sqrt{\text{\# of repeats}}$.

### 4.3 Object Detection

In Table 2, we select several downstream tasks from various domains and compare AutoMM with baseline frameworks from both Vertex AI[2] and NVIDIA TAO[3]. **Throughout the comparison, we demonstrate that AutoMM surpasses other AutoML object detection solutions in terms of performance and speed while offering greater ease of use**. While Vertex AI provides efficient model development and deployment tools, it may have limitations in pricing flexibility and customization options compared to other solutions. In our experiments, due to the high cost of Vertex AI, we focused on evaluating it using the "Higher accuracy (new)" option on only four datasets. In contrast, we conducted a thorough comparison with NVIDIA TAO using a pretrained FAN-L-Hybrid backbone [Zhou et al., 2022]. It is important to note that both Vertex AI and NVIDIA TAO have specific data requirements, necessitating additional preprocessing beyond the common COCO [Lin et al., 2014] format. Furthermore, NVIDIA TAO requires configuration for each dataset. In contrast, AutoMM can be used with just a few lines of code without additional data processing or configuration. For further details on the datasets, baseline methods, metrics, and additional setup information, please refer to the appendix.

### 4.4 Semantic Segmentation

In Table 3, we compare AutoMM with other open-source semantic segmentation toolboxes across datasets from diverse domains. AutoMM demonstrates superior or comparable performance **with minimal trainable model parameters**. This is achieved through SAM's parameter-efficient fine-tuning, enabling effective segmentation results within a low parameter budget. Additionally, compared to these toolboxes, AutoMM offers a more streamlined approach to training setup as outlined in Section 3.2, eliminating the need of rewriting dataloaders or configuration files for new datasets. For further details on datasets, baseline methods, metrics, and additional setup information, please refer to the appendix.

## 5 Conclusion and Future Work

This paper introduces AutoMM, an AutoML toolbox with a focus on foundation models and multimodal learning. Key of AutoMM lies in its support for multiple modalities, tasks, and model zoos, achieved through a unified internal pipeline and user-friendly APIs. To evaluate AutoMM's unique capacities, we also build comprehensive benchmarks covering unimodal and multimodal data, as well as basic and advanced tasks. Experimental results demonstrate AutoMM's superior performance and ease of use. While AutoMM represents an initial step towards bridging practical

---

[2]Vertex AI, Google Cloud, `https://cloud.google.com/vertex-ai`.
[3]NVIDIA TAO Toolkit, NVIDIA Developer, `https://developer.nvidia.com/tao-toolkit`.

| Method | #Params (M) | Medical | | | Natural Images | | | Agriculture | Remote Sensing |
|--------|-------------|---------|---|---|----------------|---|---|-------------|----------------|
| | | Kvasir | CVC-612 | ISIC 2017 | CAMO | SBU | Trans10K-v2 | Leaf | Road |
| | | $S_\alpha \uparrow E_\phi \uparrow$ | $S_\alpha \uparrow E_\phi \uparrow$ | Jac $\uparrow$ | $S_\alpha \uparrow E_\phi \uparrow$ | BER $\downarrow$ | mIoU $\uparrow$ | IoU $\uparrow$ | IoU $\uparrow$ |
| Detectron2 | 47.56 | 90.4 94.5 | 89.6 91.8 | 76.1 | 73.4 81.7 | 7.11 | **70.8** | 66.6 | 54.9 |
| OpenSeg | 74.50 | **92.2 95.4** | **93.3 95.3** | **78.2** | 76.3 81.1 | 7.92 | 66.1 | **78.9** | 35.3 |
| AutoMM | 8.80 | 92.1 94.7 | 90.5 92.5 | 77.9 | **89.3 92.9** | **3.90** | 69.2 | 72.9 | **62.0** |

Table 3: Comparison of AutoMM and baselines on semantic segmentation tasks in various domains.

AutoML development with cutting-edge AI research, closing the remaining gap requires significant effort. AutoMM is in active development, and several potential directions are being considered:

**Multimodal Foundation Models**. The current late-fusion model can accommodate various modality combinations, but its performance may be suboptimal because unimodal foundation models are pretrained independently. Using multimodal foundation models that capture modality interactions during pretraining could enhance performance on some downstream tasks.

**Generative Tasks**. While AutoMM supports discriminative tasks, e.g., classification and regression, there is a need to include generative tasks like image generation and question answering. Given the growing interest in generative AI and the availability of open-source generative foundation models, incorporating support for generative tasks could expand AutoML capabilities significantly.

**More Modalities**. Many real-world applications involve data modalities beyond images, text, and tabular data. AI research has seen rapid progress in modalities such as documents, audio, and video. Expanding AutoMM to support these modalities and integrating relevant research outcomes can further broaden the scope of AutoML.

## 6 Broader Impact Statement

AutoMM presents a notable advancement in AutoML by providing a unified framework capable of processing multimodal data for complex tasks with minimal input required. It democratizes machine learning, enabling non-experts to utilize advanced capabilities for tasks like object detection and image segmentation. This can spur innovation and enhance productivity across multiple domains.

The potential negative impacts of the proposed approach are similar to those of other methods reliant on foundation models, encompassing issues such as data privacy, security concerns, and the perpetuation of bias in machine learning models. Like these methods, AutoMM's effectiveness and ethical implications are tightly coupled with the characteristics of the underlying data and the design of the foundation models it employs. It is recommended that users must scrutinize their data to prevent the reinforcement of existing biases through the models trained on them.

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

## A  Reproducibility

We provide comprehensive instructions and scripts to reproduce this paper's results in the following Github repository: `https://github.com/tonyhoo/automm-benchmark-paper`. Within this repository, you can find the raw data results as well. Moreover, AutoMM offers a tutorial website (`https://auto.gluon.ai/stable/tutorials/multimodal/index.html`) hosting many hands-on tutorials such as:

- Image prediction (`https://auto.gluon.ai/stable/tutorials/multimodal/image_prediction/beginner_image_cls.html`)

- Text prediction (`https://auto.gluon.ai/stable/tutorials/multimodal/text_prediction/beginner_text.html`)

- Multimodal prediction (`https://auto.gluon.ai/stable/tutorials/multimodal/multimodal_prediction/beginner_multimodal.html`)

- Image-to-image semantic matching (`https://auto.gluon.ai/stable/tutorials/multimodal/semantic_matching/image2image_matching.html`)

- Text-to-text semantic matching (`https://auto.gluon.ai/stable/tutorials/multimodal/semantic_matching/text2text_matching.html`)

- Image–text semantic matching (`https://auto.gluon.ai/stable/tutorials/multimodal/semantic_matching/image_text_matching.html`)

- Object detection (`https://auto.gluon.ai/stable/tutorials/multimodal/object_detection/quick_start/quick_start_coco.html`)

- Semantic segmentation (`https://auto.gluon.ai/stable/tutorials/multimodal/image_segmentation/beginner_semantic_seg.html`)

Each tutorial includes both toy data and explanatory code. Additionally, a Colab link is provided at the beginning of each tutorial, enabling users to execute the tutorial while following along. When running AutoMM tutorials in Colab, it's important to select the GPU accelerator. Note that achieving complete reproducibility across different environments can be challenging due to the numerous dependencies AutoMM relies on. As highlighted in our setup.py file (https://github.com/autogluon/autogluon/blob/master/multimodal/setup.py#L23-L56), ensuring the reproducibility of each dependency is not always feasible. For instance, PyTorch, one of our crucial dependencies, acknowledges the difficulty in guaranteeing reproducibility across various releases, commits, or platforms. More information on this can be found in their documentation (https://pytorch.org/docs/stable/notes/randomness.html#reproducibility).

## B  Tutorials

This section presents an exhaustive list of our hands-on tutorials, with links directing to our official documentation website. Each tutorial comes with a short introduction for better readability.

### B.1  Text Data - Classification/Regression/NER

- AutoMM for Text Prediction - Quick Start. How to quickly train text prediction models with AutoMM.

- AutoMM for Text Prediction - Multilingual Problems. How to use AutoMM to build models for non-English languages.

- AutoMM for Named Entity Recognition - Quick Start. How to use AutoMM for entity extraction from text data.

## B.2 Image Data – Classification / Regression

- [AutoMM for Image Classification - Quick Start.](#) How to quickly train image classification models with AutoMM.

- [AutoMM for Zero-shot Image Classification.](#) How to enable zero-shot image classification in AutoMM via pretrained CLIP.

## B.3 Multimodal Data – Classification / Regression / NER

- [AutoMM for Image+Text+Tabular - Quick Start.](#) How to use AutoMM to train a model on image, text, categorical, and numeric data.

- [AutoMM for Text+Tabular - Quick Start.](#) How to apply AutoMM to multimodal data tables with a mix of text, categorical, and numeric columns.

- [AutoMM for Entity Extraction with Text and Image - Quick Start.](#) How to use AutoMM to train a model for multimodal named entity recognition.

## B.4 Image/Text Data – Semantic Matching

- [Text-to-text Semantic Matching with AutoMM - Quick Start.](#) How to use AutoMM to train models for measuring the similarity of two text items.

- [Image-to-image Semantic Matching with AutoMM - Quick Start.](#) How to use AutoMM to train models for measuring the similarity of two images.

- [Image-Text Semantic Matching with AutoMM - Quick Start.](#) How to use AutoMM to train models for matching image and text data.

- [Image-Text Semantic Matching with AutoMM - Zero-shot.](#) How to use AutoMM for zero-shot matching of image and text data.

- [Text Semantic Search with AutoMM.](#) How to use semantic embeddings to improve search ranking performance.

## B.5 Image Data – Object Detection

- [AutoMM for Object Detection - COCO Format.](#) How to use AutoMM to quickly train a object detector on a dataset with the COCO format.

- [AutoMM for Object Detection - DataFrame Format.](#) How to use AutoMM to quickly train a detector with the data in the DataFrame format.

- [Prepare COCO2017 Dataset.](#) How to prepare COCO2017 dataset for object detection.

- [Prepare Pascal VOC Dataset.](#) How to prepare Pascal VOC dataset for object detection.

- [Prepare Watercolor Dataset.](#) How to prepare Watercolor dataset for object detection.

- [Prepare Pathhole Dataset.](#) How to prepare Pathhole dataset for object detection.

- [Convert VOC Format Dataset to COCO Format.](#) How to convert a dataset from the VOC format to the COCO format.

- [Finetune on COCO Format Dataset with Customized Settings.](#) How to quickly customize high quality object detection model with AutoMM on COCO format datasets.

### B.6 Image Data – Segmentation

- [AutoMM for Semantic Segmentation - Quick Start.](#) How quickly train semantic segmentation models with AutoMM.

### B.7 Advanced Topics

- [Single GPU Billion-scale Model Training via Parameter-Efficient Finetuning.](#) How to take advantage of large foundation models with the help of parameter-efficient finetuning.

- [Hyperparameter Optimization in AutoMM.](#) How to do hyperparameter optimization in AutoMM.

- [Knowledge Distillation in AutoMM](#) How to do knowledge distillation in AutoMM.

- [Customize AutoMM.](#) How to customize AutoMM configurations.

- [AutoMM Presets.](#) How to use AutoMM presets.

- [Few Shot Learning with AutoMM.](#) How to use foundation models + SVM for few shot learning.

- [Handling Class Imbalance with AutoMM - Focal Loss.](#) How to use AutoMM to handle class imbalance.

- [Faster Prediction with TensorRT.](#) How to use TensorRT in accelerating AutoMM model inference.

- [Continuous Training with AutoMM.](#) Different use cases for continuous training with AutoMM.

## C Presets

In the experiments, AutoMM used the `best_quality` preset. Given a problem type, the hyperparameters are fixed, which means different datasets of the same problem type use the same hyperparamters without tuning. The detailed hyperparameters of each problem type are provided in Table 4.

The presets hyperparameters were determined through an offline search on our internal benchmark datasets. During this searching, we conducted random search for each task on the combinations of model backbone (task-specific model pools), feature pooling mode (CLS, mean), batch size (4, 8, 16, 32, 64, 128, 256), learning rate (1e-5, 1e-4, 1e-3), learning rate choice (layerwise decay, two-stage, single-stage), weight decay (1e-2, 1e-3, 1e-4), learning rate schedule (cosine decay, multi-step, polynomial decay, no decay), warmup steps (0.1, 0.2, 0.3), patience (5, 10, 20), validation check internal (0.5, 1), and epochs (10, 20, 30, 40, 50, 60). We used the budget of running 2000 jobs per task.

Based on the searched presets, we find that different tasks generally favor different foundation models. For example, the basic task uses deberta-v3 as the text backbone, but the text-text matching uses all-mpnet-base-v2. Most preset hyperparameters of object detection and semantic segmentation are different from those the basic classfication/regression. For instance, the basic task uses layerwise leaning rate decay, object detection uses two-stage learning rate, and semantic segmentation utilizes a single-stage learning rate. While the presets of semantic matching are similar to the basic task, the learning rate of image-text matching is different from the basic task's (1e-5 vs. 1e-4).

## D Classification and Regression

### D.1 Dataset Details

We have included datasets of a wide range of domains to evaluate on the classification and regression tasks, and summarized in Table 5.

**Text Classification and Natural Language Understanding**. We include datasets like `MLDoc-11000`[Schwenk and Li, 2018] and `ag_news`[Zhang et al., 2015] for document and news

| | Multimodal Classification/Regression (best_quality) | Semantic Matching TTM (best_quality) | Semantic Matching IIM (best_quality) | Semantic Matching ITM (best_quality) | Object Detection (best_quality) | Semantic Segmentation (best_quality) |
|---|---|---|---|---|---|---|
| text model | deberta-v3-base | \ | \ | \ | \ | \ |
| image model | swin_large | all-mpnet-base-v2 | swin_large | \ | \ | \ |
| tabular model | ft_transformer | \ | \ | \ | \ | \ |
| task specific model | \ | \ | \ | clip-vit-L14-336 | dino-5scale_swin-l | sam-vit-huge |
| pooling_mode | cls | mean | \ | \ | \ | \ |
| batch_size | 128 | 128 | 128 | 128 | 32 | 4 |
| precision | 16-mixed | 16-mixed | 16-mixed | 16-mixed | 16-mixed | 16-mixed |
| learning_rate | 1e-4 | 1e-4 | 1e-4 | 1e-5 | 1e-4 | 1e-4 |
| lr_choice | layerwise_decay | layerwise_decay | layerwise_decay | layerwise_decay | two-stage | single_stage |
| layerwise_lr_decay | 0.9 | 0.9 | 0.9 | 0.9 | \ | \ |
| low_lr_layers | \ | \ | \ | \ | ["backbone"] | \ |
| lr_multiplier | \ | \ | \ | \ | 0.1 | \ |
| weight_decay | 0.001 | 0.001 | 0.001 | 0.001 | 0.0001 | 0.0001 |
| gradient_clip_val | 1 | 1 | 1 | 1 | 0.1 | 1 |
| lr_schedule | cosine_decay | cosine_decay | cosine_decay | cosine_decay | multi-step | polynomial_decay |
| lr_steps | \ | \ | \ | \ | [30, 55] | \ |
| warmup_steps | 0.1 | 0.1 | 0.1 | 0.1 | 0. | 0. |
| patience | 10 | 10 | 10 | 10 | 20 | 10 |
| val_check_interval | 0.5 | 0.5 | 0.5 | 0.5 | 1.0 | 1.0 |
| check_val_every_n_epoch | 1 | 1 | 1 | 1 | 1 | 1 |
| max_epochs | 10 | 10 | 10 | 10 | 60 | 30 |
| efficient_finetune | \ | \ | \ | \ | \ | lora (r=32,a=32) |

Table 4: Preset configuration of AutoMM for each problem type.

| Dataset | Domain | Problem Type | #Train | #Val | #Test | #Category | Task Description |
|---|---|---|---|---|---|---|---|
| fashion_moist | Fashion | Multiclass | 60000 | 0 | 10000 | 10 | Identify fashion product categories |
| food101 | Food | Multiclass | 75750 | 0 | 25250 | 101 | Identify food categories |
| Stanford_cars | Automotive | Multiclass | 8144 | 0 | 8041 | 196 | Identify car models |
| magnetic_tile_defects | Industrial | Multiclass | 1008 | 0 | 336 | 6 | Identify defects in tiles |
| European_flood_depth | Environmental Science | Binary | 3153 | 0 | 557 | 2 | Identify flood types |
| Oxford_flowers | Botany | Multiclass | 1020 | 0 | 6149 | 102 | Identify flower types |
| OxfordIIITPet | Veterinary | Multiclass | 4436 | 1478 | 1479 | 37 | Identify pet breeds |
| CD18_cellphone | Consumer Product | Regression | 2532 | 0 | 633 | NA | Predict cellphone price |
| HAM10000 | Medical | Multiclass | 10015 | 0 | 1512 | 7 | Identify dermatological disease types |
| hateful_meme | Social Media | Binary | 6800 | 0 | 1700 | 2 | Detect harmful content |
| petfinder | Animal Welfare | Multiclass | 11994 | 0 | 2999 | 5 | Predict adoption speed |
| memotion | Social Media | Multiclass | 5593 | 0 | 1399 | 5 | Categorize sentiment |
| financial_news | Media | Multiclass | 3876 | 0 | 969 | 3 | Categorize sentiment |
| MLDoc-11000 | Information Retrieval | Multiclass | 9777 | 1223 | 4000 | 4 | Document classification |
| gnad10 | Social Media | Multiclass | 8228 | 1017 | 1028 | 9 | German news articles categorization |
| MultiATIS-5000 | Travel | Multiclass | 4424 | 554 | 891 | 17 | Intent recognition |
| fb_dialog | Social Media | Multiclass | 25288 | 3162 | 7799 | 12 | Intent recognition |
| SNIPS | Technology | Multiclass | 13084 | 700 | 700 | 7 | Intent recognition |
| ag_news | Media | Multiclass | 106666 | 13334 | 7600 | 4 | Identify news topic |
| airbnb_melbourn | Real Estate | Multiclass | 18316 | 0 | 4579 | 10 | Predict price label |
| kick_start_funding | Business | Binary | 86502 | 0 | 21626 | 2 | Predict crowdfunding campaign's success |
| cloth_review | Fashion | Multiclass | 18788 | 0 | 4698 | 5 | Categorize sentiment |
| news_popularity | Media | Regression | 24007 | 0 | 6002 | NA | Predict the number of shares of news articles |
| California_house | Real Estate | Regression | 37951 | 0 | 9488 | NA | Predict house prices |

Table 5: Basic Task Datasets

classification tasks. `MLDoc-11000` is geared towards multilingual document classification, whereas `ag_news` focuses on categorizing news articles. The `MultiATIS-5000`[Xu et al., 2020] dataset is used for understanding user input in the context of the Air Travel Information System. `SNIPS`[Coucke et al., 2018] is another dataset aimed at natural language understanding, tailored for an AI assistant's conversational understanding. The `cloth_review`[Agarap, 2018] dataset provides a basis for sentiment analysis in customer reviews for clothing items, and the `financial_news`[Malo et al., 2014] dataset is used for sentiment analysis within the financial domain.

**Image Classification**. For visual recognition tasks, we employ datasets like `OxfordIIITPet`[Parkhi et al., 2012], `fashion_mnist`[Xiao et al., 2017], `food101`[Bossard et al., 2014], and `stanfordcars`[Krause et al., 2013]. `OxfordIIITPet`[Parkhi et al., 2012] contains images for pet breed identification, `fashion_mnist` for clothing articles, `food101` for various food categories, and `stanfordcars` for automobile models. These datasets vary in complexity and are standard benchmarks in the field of computer vision.

**E-Commerce and Housing**. The `airbnb`[air] dataset is included for predicting housing prices and could be used for recommendation systems within the housing rental market. Similarly, the `cal_house`[Pace and Barry, 1997] dataset provides data for California housing price prediction, which is a classic regression problem.

**Medical Imagery**. The `ham10000`[Tschandl et al., 2018] dataset is integral for medical image analysis, focusing on skin lesion classification for disease detection.

**Defect Detection**. We utilize `magnetictiledefects`[Huang et al., 2020] dataset for defect detection in manufacturing processes. This dataset is used for automating quality control in an industrial setting.

**Sentiment Analysis and Memetics**. The `hateful_meme`[Kiela et al., 2021] and `memotion`[Mishra et al., 2023] datasets are curated for detecting sentiment and emotion in memes, which is an emerging field in natural language processing and computer vision.

**Others**. Additional datasets such as `petfinder`[pet] are used for pet adoption services, and `news_popularity`[Fernandes et al., 2015] for predicting the popularity of news articles, demonstrating the varied applications of machine learning across different sectors.

## D.2 Baseline Methods

We considered several other AutoML frameworks as baselines, e.g. Auto-Sklearn, Auto-Keras, and H2O AutoML. However, among these, only Auto-Keras supports multimodal input.

## D.3 Metrics

The metrics being used depend on the specific tasks.

For regression problems with continuous labels, we use the coefficient of determination, commonly known as R-squared ($R^2$):

$$R^2 = 1 - \frac{\sum_{i=1}^{n}(y_i - \hat{y}_i)^2}{\sum_{i=1}^{n}(y_i - \bar{y})^2},$$

where $y_i$ is the value of the label, $\hat{y}_i$ is the predicted value from the regression model, and $\bar{y}$ is the mean of the actual values $y_i$ over all $n$ observations.

For binary classification tasks, we use the F1 score ($f1$), which is the harmonic mean of precision and recall:

$$\text{F1} = 2 \times \frac{\text{precision} \times \text{recall}}{\text{precision} + \text{recall}},$$

where

$$\text{precision} = \frac{\text{True Positive}}{\text{True Positive} + \text{False Positive}} \quad \text{and} \quad \text{recall} = \frac{\text{True Positive}}{\text{True Positive} + \text{False Positive}}$$

.

For multiclass classification tasks, we use the weighted F1 score ($f1_{\text{weighted}}$), which is a weighted average of the F1 score for each class, taking into account label imbalance:

$$F1_{\text{weighted}} = \sum_{i=1}^{C} w_i \cdot f1_i$$

where $C$ is the number of classes, $F1_i$ is the F1 score for class $i$, $w_i$ is the proportion of true instances for class $i$ in the dataset.

All reported metrics are computed on a holdout test set not used for training or hyperparameter selection. Some of the datasets include a test set while others do not. In the cases where they do not, we randomly select 20% of the training data as the test set.

### D.4 Setup

We used AutoMM's `best_quality` presets for all datasets. Since there is no presets provided in Auto-Keras, we adopted the default configuration as suggested by their tutorial.

We used AWS EC2 g4dn.12xlarge instances equipped with 4 NVIDIA T4 GPUs for all experiment runs. The total GPU time used for training is about 852 hours, which includes 5 repeats with ablation studies on 24 datasets.

### D.5 Ablation Studies

Given one pre-trained model, different fine-tuning techniques may result in performance variance. To tease apart the effects of various tricks used in fine-tuning, we conducted extensive ablation studies. AutoMM `best_quality` preset have 5 tricks applied (*cosine_decay*, *gradient_clip*, *greedy_soup*, *layerwise_lr_decay* and *weight_decay*). We conducted the experiment by adding each of the 5 tricks to the baseline separately with results presented in columns "*+ trick_name*", and also by applying all tricks to the baseline ("*+ all*"). The results are shown in Table 6. Our findings indicate significant performance improvements when employing greedy soup ("+ greedy_soup") or incorporating all fine-tuning enhancements ("+ all") compared to the baseline model without any modifications ("AutoMM_base"). Notably, the highest win-rate observed compared across all configurations in our study was achieved with the "+ all" configuration, recording a win-rate of 0.625 (15/24), which forms the basis for our "best_quality" preset.

## E  Object Detection

### E.1  Dataset Details

Following [Fang et al., 2024], we choose to evaluate the performance on several downstream object detection datasets spanning diverse domains such as agriculture, medical, comic domain transfer, remote sensing, autonomous driving, and infrared imagery. We have excluded simpler datasets while incorporating larger and more intricate ones to ensure a comprehensive evaluation. These datasets are summarized in Table 7.

**Agriculture**. We choose plantdoc [Singh et al., 2020] dataset for plant disease detection and deepfruits [Sa et al., 2016] dataset for fruits detection. Plantdoc has 31 categories, containing 2328 images with 8353 bounding boxes for training and validation, and 239 images with 454 bounding boxes for testing. Deepfruits has 7 categories, containing 457 images with 2552 bounding boxes for training and validation, and 114 images with 589 bounding boxes for testing.

**Medical Images**. We choose ChestX-Det10 [Liu et al., 2020] dataset for detection of Thoracic Abnormalities in chest X-ray and deeplesion [Yan et al., 2018] dataset for lesions detection in CT images. ChestX-Det10 has 10 categories, containing 2320 images with 6863 bounding boxes for training and validation, and 459 images with 1476 bounding boxes for testing. Deeplesion has 1

| Task | AutoMM_base | + cosine_decay | + grad_clip | + greedy_soup | + layerwise_lr_decay | + weight_decay | + all |
|---|---|---|---|---|---|---|---|
| fashion_mnist ↑ | 0.947(0.002) | 0.947(0.002) | 0.947(0.002) | **0.955(0.001)** | 0.948(0.001) | 0.946(0.003) | 0.953(0.002) |
| food101 ↑ | 0.920(0.003) | 0.921(0.001) | 0.917(0.002) | 0.935(0.002) | 0.923(0.003) | 0.917(0.003) | **0.937(0.001)** |
| stanfordcars ↑ | 0.877(0.003) | 0.879(0.003) | 0.871(0.007) | 0.890(0.004) | 0.881(0.003) | 0.879(0.003) | **0.892(0.002)** |
| magnetictiledefects ↑ | 0.955(0.010) | **0.964(0.008)** | 0.956(0.016) | 0.961(0.007) | 0.959(0.010) | 0.961(0.008) | 0.956(0.014) |
| europeanflooddepth ↑ | 0.785(0.017) | 0.789(0.014) | 0.786(0.014) | **0.796(0.005)** | 0.786(0.010) | 0.788(0.007) | 0.790(0.008) |
| oxfordflowers ↑ | 0.991(0.001) | 0.990(0.002) | 0.991(0.003) | 0.989(0.004) | 0.991(0.001) | **0.992(0.002)** | 0.989(0.003) |
| OxfordIIITPet ↑ | 0.955(0.005) | 0.956(0.003) | 0.952(0.003) | **0.960(0.002)** | 0.958(0.006) | 0.956(0.002) | 0.958(0.003) |
| cd18 ↑ | -0.037(0.506) | -2.209(4.705) | -1.368(3.740) | **-0.033(0.507)** | -1.649(3.426) | -1.827(4.392) | -1.843(4.477) |
| ham10000 ↑ | 0.532(0.042) | 0.563(0.023) | 0.594(0.024) | 0.538(0.027) | 0.588(0.017) | 0.569(0.020) | **0.608(0.014)** |
| hateful_meme ↑ | 0.577(0.018) | 0.589(0.027) | **0.612(0.023)** | 0.590(0.020) | 0.586(0.013) | 0.599(0.012) | 0.596(0.013) |
| petfinder ↑ | 0.398(0.008) | 0.398(0.006) | 0.402(0.007) | 0.397(0.008) | 0.403(0.006) | 0.389(0.011) | **0.408(0.006)** |
| memotion ↑ | 0.446(0.014) | 0.455(0.014) | 0.458(0.021) | 0.449(0.018) | 0.457(0.014) | 0.457(0.020) | **0.467(0.013)** |
| financial_news ↑ | 0.868(0.011) | 0.866(0.013) | 0.863(0.004) | 0.870(0.012) | 0.868(0.011) | 0.868(0.006) | **0.874(0.010)** |
| MLDoc-11000 ↑ | 0.974(0.002) | 0.973(0.003) | 0.973(0.002) | 0.977(0.001) | 0.975(0.002) | 0.973(0.002) | **0.978(0.002)** |
| gnad10 ↑ | 0.881(0.006) | 0.874(0.007) | 0.880(0.004) | 0.889(0.006) | 0.891(0.004) | 0.880(0.008) | **0.899(0.006)** |
| MultiATIS-5000 ↑ | 0.989(0.001) | 0.989(0.002) | **0.990(0.001)** | 0.990(0.001) | 0.987(0.004) | 0.987(0.005) | 0.990(0.003) |
| fb_dialog ↑ | 0.992(0.001) | 0.992(0.001) | 0.991(0.001) | **0.993(0.000)** | 0.992(0.001) | 0.991(0.001) | 0.992(0.001) |
| SNIPS ↑ | 0.989(0.003) | 0.986(0.003) | 0.985(0.001) | 0.988(0.003) | 0.987(0.005) | 0.989(0.002) | **0.990(0.002)** |
| ag_news ↑ | 0.935(0.002) | 0.938(0.002) | 0.938(0.001) | 0.941(0.002) | 0.941(0.002) | 0.937(0.001) | **0.944(0.001)** |
| airbnb ↑ | 0.313(0.008) | 0.309(0.007) | 0.323(0.020) | 0.314(0.009) | 0.342(0.016) | 0.305(0.012) | **0.397(0.011)** |
| kick_start ↑ | 0.331(0.166) | 0.383(0.199) | 0.296(0.255) | 0.415(0.213) | 0.590(0.009) | 0.279(0.243) | **0.609(0.005)** |
| cloth_review ↑ | 0.719(0.010) | 0.717(0.007) | 0.722(0.002) | 0.725(0.012) | 0.730(0.005) | 0.716(0.008) | **0.735(0.004)** |
| news_popularity ↑ | 0.008(0.001) | 0.009(0.003) | 0.008(0.001) | 0.009(0.002) | 0.012(0.001) | 0.009(0.003) | **0.014(0.003)** |
| cal_house ↑ | 0.929(0.008) | 0.931(0.002) | 0.937(0.002) | 0.929(0.003) | 0.938(0.001) | 0.929(0.004) | **0.944(0.001)** |

Table 6: Ablation studies for the classification and regression experiments. The numbers indicate the average performance metric (and error bars) for each dataset as we introduce each option. The error bars are estimated using 5 independent repeats with different random seeds using $1.96 \times \text{std}/\sqrt{\text{\# of repeats}}$.

| Dataset | Domain | #Train | #Val | #Test | #Category | Task Description |
|---|---|---|---|---|---|---|
| plantdoc | Agriculture | 8353 | 454 | 457 | 31 | Detection of plant desease detection. |
| deepfruits | | 2552 | 0 | 589 | 7 | Detection of fruits detection. |
| chest10 | Medical | 6863 | 0 | 1476 | 10 | Detection of Thoracic Abnormalities in chest X-ray. |
| deeplesion | | 23785 | 5085 | 5121 | 1 | Detection of lesions in CT images. |
| comic | Domain Transfer | 3213 | 0 | 3175 | 6 | Detection of common objects in comic domain. |
| dota | Remote Sencing | 181746 | 54426 | 236172 | 15 | Detection of objects in aerial images. |
| kitti | Autonomous Driving | 38076 | 0 | 52457 | 3 | Detection in self driving scenario. |
| thermal | Infrared | 181 | 49 | 27 | 3 | Detection for infrared images. |

Table 7: Object Detection Datasets

categories, containing 22919 images with 23785 bounding boxes for training, 4889 images with 5085 bounding boxes for validation, and 4927 images with 5121 bounding boxes for testing.

**Comic Domain Transfer**. We choose comic [Inoue et al., 2018] dataset for detection of common objects in comic domain. Comic has 6 categories, containing 1000 images with 3213 bounding boxes for training and validation, and 1000 images with 3175 bounding boxes for testing.

**Remote Sensing**. We choose dota [Xia et al., 2018] dataset for for object detection in aerial images. Dota has 15 categories, containing 9305 images with 181746 bounding boxes for training and validation, and 2955 images with 54426 bounding boxes for testing.

**Autonomous Driving**. We choose KITTI [Geiger et al., 2012] dataset for for object detection in the self driving scenario. KITTI has 3 categories, containing 5481 images with 38076 bounding boxes for training and validation, and 7481 images with 52457 bounding boxes for testing.

**Infrared Imagery**. We choose Thermal Dogs and People [Ciaglia et al., 2022] dataset for for object detection for infrared images. Thermal Dogs and People dataset has 3 categories, containing 142 images with 181 bounding boxes for training, 41 images with 49 bounding boxes for validation, and 20 images with 27 bounding boxes for testing.

| Dataset | Domain | #Train | #Val | #Test | Task Description |
|---------|--------|--------|------|-------|------------------|
| Kvasir | Medical | 720 | 180 | 100 | segment abnormal growths within gastrointestinal endoscopic images. |
| CVC-612 | Medical | 440 | 110 | 62 | segment abnormal growths within gastrointestinal endoscopic images. |
| ISIC2017 | Medical | 200 | 600 | 150 | segment various types of skin lesions. |
| CAMO | Natural Images | 3636 | 404 | 250 | segment objects that are concealed within complex backgrounds. |
| SBU | Natural Images | 3677 | 408 | 638 | recognize shadow regions within a scene. |
| Trans10K-v2 | Natural Images | 5000 | 1000 | 4428 | multi-class transparent object segmentation. |
| Leaf | Agriculture | 399 | 99 | 90 | segment individual plant leaf diseases within agricultural images. |
| Road | Remote Sensing | 1107 | 13 | 48 | segment road or street regions within images or video frames. |

Table 8: Semantic Segmentation Datasets

## E.2 Baseline Methods

We compare AutoMM with baseline frameworks from both Vertex AI and Nvidia TAO. While Vertex AI offers efficient model development and deployment tools, it may have limitations in pricing flexibility and customization options compared to other solutions. In our experiment, we focused on evaluating Vertex AI only on four datasets using its "Higher accuracy (new)" option due to its high cost, while conducting a thorough comparison with Nvidia TAO using DINO [Zhang et al., 2022] with a pretrained FAN-L-Hybrid backbone [Zhou et al., 2022] for 60 epochs training. It's important to note that both Vertex AI and Nvidia TAO have specific data requirements, requiring additional preprocessing beyond the common COCO[Lin et al., 2014] format, while Nvidia TAO additionally requires configuration per dataset.

## E.3 Metrics

For object detection evaluation, mAP [Lin et al., 2014] and AP50 [Lin et al., 2014] are the key metrics used for assessing both Nvidia Tao and AutoMM models. However, in the case of Vertex AI, only AP50 is reported as the evaluation metric in their service.

## E.4 Raw Results

We provide the model performance in Table 2. However, due to the considerable time and computational resources required, we were unable to conduct multiple rounds of experiments for object detection tasks.

## E.5 Setup

We use a AWS EC2 P4d.24xlarge server with 8x A100 40G GPUs for training and evaluating Nvidia TAO and AutoMM. For Vertex AI, we use their online service API.

## F  Semantic Segmentation

## F.1 Dataset Details

Following [Zhong et al., 2024], we choose several datasets from real-world semantic segmentation scenarios including medical, natural images, agriculture and remote sensing. These datasets are summarized in Table 8.

**Medical Images**. We choose two polyp segmentation datasets Kvasir [Jha et al., 2020] and CVC-612 [Bernal et al., 2015], and one skin lesion segmentation dataset ISIC 2017 [Codella et al., 2018]. Kvasir contains 1000 images and CVC-612 includes 612 open-access images. We randomly divide a validation set comprising 20% of the images from the training set, for validation during training. ISIC 2017 provides 2000 images for training, 150 images for 643 validation and 600 images for testing.

**Natural Images**. We choose CAMO [Le et al., 2019] for camouflaged object segmentation, SBU [Vicente et al., 2016] for shadow detection, and Trans10K-v2 [Xie et al., 2021] dataset for

| Method | Medical | | | Natural Images | | | Agriculture | Remote Sensing |
|---|---|---|---|---|---|---|---|---|
| | Kvasir | CVC-612 | ISIC 2017 | CAMO | SBU | Trans10K-v2 | Leaf | Road |
| Detectron2 | 3.3 | 3.3 | 10.8 | 13.7 | 11.3 | 15.6 | 1.1 | 4.4 |
| OpenSeg | 3.2 | 3.2 | 6.6 | 9.4 | 9.1 | 15.0 | 1.3 | 2.7 |
| AutoMM | 5.2 | 5.2 | 8.3 | 15.3 | 15.7 | 22.2 | 1.5 | 5.1 |

Table 9: Training time (hours) of semantic segmentation experiments.

multi-class transparent object segmentation. CAMO provides 1000 images for training and 250 for testing. We train on the combined dataset consists of the training images from COD10K [Fan et al., 2020] and CAMO for 20 epochs, and test on the three datasets. Additionally, we randomly split 10% of the images from the training set for validation. SBU contains 4085 and 638 images for training and testing. We randomly split 10% of the images from the training set for validation. Trans10K-v2 contains background plus two main categories divided into 11 fine-grained categories, using 5000, 1000 and 4428 images for training, validation and testing, respectively.

**Agriculture**. We use Leaf [Rath, 2023] dataset for leaf disease segmentation. It contains 498 images for training and 90 images for testing. We randomly split the training images into 80% for training, 20% for validation.

**Remote Sensing**. We choose Massachusetts Roads Dataset [Mnih, 2013] for road segmentation, which contains 1107 images for training, 13 images for validation and 48 images for testing.

## F.2 Baseline Methods

**Detectron2** [Wu et al., 2019], developed by Facebook AI Research, is a cutting-edge library offering state-of-the-art detection and segmentation algorithms. We use its supported 'SemanticFPN+PointTrend' method as our baseline. PointRend [Kirillov et al., 2019] can be seamlessly integrated into instance and semantic segmentation tasks atop existing state-of-the-art models. And 'SemanticFPN+PointTrend' demonstrates superior performance on the Cityscapes [Cordts et al., 2016] semantic segmentation dataset.

**OpenSeg** [Contributors, 2019] is the official PyTorch implementation of the OCNet [Yuan and Wang, 2018] series and SegFix [Yuan et al., 2020]. We choose its supported 'OCR+RMI' method, recognized as the state-of-the-art in their benchmark study. This method employs HRNet [Sun et al., 2019] as its backbone, ensuring the preservation of high-resolution representations throughout both the image encoding and decoding stages. Moreover, the training loss RMI loss [Zhao et al., 2019], effectively utilizes region mutual information (RMI) to model dependencies among pixels.

## F.3 Metrics

We use $S_\alpha$ [Fan et al., 2017] and $E_\phi$ [Fan et al., 2018] as our metrics for polyp segmentation and camouflaged object segmentation. These metrics are widely acknowledged within these domains. $S_\alpha$ quantifies the similarity between predictions and ground-truths, while $E_\phi$ provides assessments at both pixel and global levels of similarity. We use Jaccard Index [Codella et al., 2018] for skin lesion segmentation as the ISBI 2017 challenge ranked methods according to it. We use Balanced Error Rate (BER) [Vicente et al., 2016] for shadow detection, which is a common metric in this area where shadow pixels are considerably less than non-shadow pixels. For leaf segmentation, road segmentation and transparent object detection, we use IoU metric.

| Dataset | #Train | #Val | #Test | Pos Ratio | Task | Metric | Task Description |
|---|---|---|---|---|---|---|---|
| MRPC | 3,261 | 815 | 1,725 | 0.67 | TTM | ROC_AUC | identify if a sentence from a newswire artile is a paraphrase of another |
| MultiNLI | 160,000 | 40,000 | 9,834 | 0.33 | TTM | ROC_AUC | identify if a sentence entails another sentence from different genres |
| Quora | 323,442 | 40,430 | 40,429 | 0.37 | TTM | ROC_AUC | identify if a Quora questions is duplicate of another question |
| SciTail | 18,870 | 4,717 | 2,126 | 0.37 | TTM | ROC_AUC | identify if a sentence entails another sentence in the science domain |
| SNLI | 293,282 | 73,321 | 6,605 | 0.50 | TTM | ROC_AUC | identify if a human-written sentence entails another sentence |
| SOP | 3,622 | 905 | 2,263 | 0.74 | IIM | ROC_AUC | identify if two images are from the same online product |
| Airbnb | 1,970 | 493 | 633 | 0.40 | IIM | ROC_AUC | identify if two images are from the same room in Airbnb |
| CUB200 | 16,151 | 2,020 | 16,515 | 0.27 | IIM | ROC_AUC | identify if two images are from the same subcategory of birds |
| iPanda50 | 4,000 | 1,000 | 2,000 | 0.30 | IIM | ROC_AUC | identify if two images are from the same panda individual |
| SeaTurtleID | 4,000 | 1,000 | 2,000 | 0.30 | IIM | ROC_ACU | identify if two images are from the same turtle individual |
| CUB200-text | 47,961 | 11,990 | 57,930 | N/A | ITM, TIM | Recall@K | retrieve the most related image/text given the query text/image |
| Flower102-text | 45,850 | 11,462 | 24,570 | N/A | ITM, TIM | Recall@K | retrieve the most related image/text given the query text/image |
| MSCOCO | 14,055 | 3,515 | 7,507 | N/A | ITM, TIM | Recall@K | retrieve the most related image/text given the query text/image |
| Flickr30K | 145,000 | 5,070 | 5,000 | N/A | ITM, TIM | Recall@K | retrieve the most related image/text given the query text/image |
| IPC | 14,579 | 2,490 | 2,492 | N/A | ITM, TIM | Recall@K | retrieve the most related image/text given the query text/image |

Table 10: Datasets used in Semantic Matching tasks, Text to Text Matching (TTM), Image to Image Matching (IIM), Text to Image Matching (TIM), Image to Text Matching (ITM). For ITM and TIM, the data only contains positive image-text pairs. For the metric Recall@K, we use K=1, 5, 10, and take the average as the final metric.

## F.4 Raw Results

We provide the model performance in Table 3. By leveraging SAM's parameter-efficient fine-tuning, our solution requires only the storage of trainable parameters for each task, unlike other toolboxes that necessitate the storage of entire trainable models for each task during fine-tuning. This enables us to alleviate storage burdens when dealing with multiple segmentation tasks in real-world scenarios, without significantly compromising performance.

Furthermore, our solution demonstrates the capacity to uphold relatively stable performance across datasets from various domains. While Detectron2 and OpenSeg outperform our method on certain datasets (e.g., Trans10K-v2) owing to their larger number of trainable parameters, our approach remains comparable within a performance margin of 6%. Notably, our method surpasses them by more than 10% on several other datasets, such as CAMO and Road.

Due to the considerable time and computational resources required, we were unable to conduct multiple rounds of experiments for semantic segmentation tasks. We run all the semantic segmentation experiments on a single NVIDIA V100 32G GPU. The total training time for each experiment is listed in Table 9.

## G Semantic Matching

### G.1 Dataset Details

Table 10 summarizes all the used semantic matching datasets. Below gives detailed descriptions of each dataset.

**MRPC (Microsoft Research Paraphrase Corpus)** [Dolan and Brockett, 2005] is a corpus consists of 5,801 sentence pairs collected from newswire articles. Each pair is labelled if it is a paraphrase or not by human annotators, which corresponds to matching or not matching in our experiments. We randomly select 20% of training data as validation data.

**MultiNLI (Multi-Genre Natural Language Inference)** [Williams et al., 2018] has 433K sentence pairs. It offers ten distinct genres (Face-to-face, Telephone, 9/11, Travel, Letters, Oxford University Press, Slate, Verbatim, Goverment and Fiction) of written and spoken English data. Each sentence pair is labelled as entailment, contradiction, or neutral. In our experiments, we label entailment as matching and contradiction and neutral as not matching. We randomly select 20% of training data as validation data.

**Quora Duplicate Questions** [Shankar et al., 2017] contains over 400,000 lines of potential question duplicate pairs. We label duplicate question pairs as matching and non-duplicate pairs as not matching.

|  | MRPC | MultiNLI | Quora | SciTail | SNLI |
|---|---|---|---|---|---|
| AutoMM | 87.37 (0.06) | 91.27 (0.04) | 95.77 (0.01) | 97.77 (0.01) | 96.71 (0.01) |
| Sentence Transformer | 88.89 (0.21) | 91.76 (0.01) | 96.33 (0.01) | 96.93 (0.04) | 96.78 (0.01) |

Table 11: Comparisons on Text to Text Matching datasets. Mean and (Vairiance) are reported.

|  | SOP | Airbnb | CUB200 | iPanda | SeaTurtle |
|---|---|---|---|---|---|
| AutoMM | 97.08 (0.01) | 99.56 (0.00) | 97.83 (0.01) | 82.43 (0.08) | 88.40 (0.33) |
| Sentence Transformer | 95.97 (0.03) | 98.57 (0.02) | 93.35 (0.01) | 90.40 (0.31) | 89.06 (0.06) |

Table 12: Comparisons on Image to Image Matching datasets. Mean and (Variance) are reported.

**SciTail** [Khot et al., 2018] dataset is an entailment dataset created from multiple-choice science exams and web sentences. Each question and the correct answer choice are converted into an assertive statement to form the hypothesis. The premise is obtained from a large text corpus of web sentences. The premise-hypothesis pairs are labelled as entails and neutral, which corresponds to matching and not matching in our experiments. The dataset contains 27,026 examples and we randomly select 20% of training data as validation data.

**SNLI (Stanford Natural Language Inference)** consists of 570K sentence-pairs manually labeled as entailment, contradiction, and neutral. In our experiments, we label entailment as matching and contradiction as not matching and discard neutral labels.

**SOP (Stanford Online Products)** [Song et al., 2016] contains 12 categories of products. Each category has some products, and each product has several images captured from different views. In the experiments, we consider different views of the same product as matching and images from different products as not matching.

**Airbnb Duplicate Image** [Airbnb, 2020] contains interior and exterior house pictures scraped from Airbnb over three cities. Each image in this dataset has at least another image which is a duplicate of the same room. We regard the images of the same room as matching samples and images of different room as not matching.

**CUB200 (Caltech-UCSD Birds-200)** [Wah et al., 2022] is the most widely-used dataset for fine-grained visual categorization task. It contains 11,788 images of 200 subcategories belonging to birds. Reed et al. [2016] further provided 10 single-sentence descriptions for each image in the dataset, which we name as CUB200-text. In our experiments, we use this dataset for Image to Image Matching, Text to Image Matching and Image to Text Matching tasks. For Image to Image Matching, we randomly select a pair of images and label them as matching pairs if they are from the same category. For Text to Image Matching and Image to Text Matching, we use the image-text pairs from Reed et al. [2016].

**iPanda-50** [Wang et al., 2021b] consists of 6,874 images of 50 giant panda individuals with 49 to 292 images per panda. The iPanda-50 dataset is used for fine-grained panda identification. In our experiments, we use it for Image to Image Matching. Similar to CUB200, we randomly select a pair of images and label them as matching pairs if they are from the same panda.

**SeaTurtleID** [Papafitsoros et al., 2022] is a public large-scale, long-span dataset with sea turtle photographs captured in the wild. It consists of 7774 images of 400 unique individuals collected within 12 years in 1081 encounters. Similar to iPanda-50, we use this dataset for Image to Image Matching, and we randomly select a pair of images and label them as matching pairs if they are from the same turtle.

**Flower102** [Nilsback and Zisserman, 2008] is a fine-grained image classification dataset consisting of 102 flower categories. Each class consists of between 40 and 258 images. Reed et al. [2016] further provided 10 single-sentence descriptions for each image in the dataset, which we name as

|                     | MSCOCO       | Flickr30K    | CUB200-text  | Flower102-text | IPC          |
|---------------------|--------------|--------------|--------------|----------------|--------------|
| AutoMM              | 82.44 (0.13) | 92.89 (0.46) | 12.75 (0.12) | 21.31 (0.01)   | 91.41 (0.11) |
| Sentence Transformer| 76.58 (0.04) | 88.31 (0.24) | 15.72 (0.01) | 24.34 (0.02)   | 90.12 (0.24) |

Table 13: Comparisons on Text to Image Matching datasets. Mean and (Vairiance) are reported.

|                     | MSCOCO       | Flickr30K    | CUB200-text  | Flower102-text | IPC          |
|---------------------|--------------|--------------|--------------|----------------|--------------|
| AutoMM              | 90.29 (0.01) | 98.26 (0.01) | 21.67 (0.24) | 31.44 (0.01)   | 91.45 (0.06) |
| Sentence Transformer| 83.13 (0.58) | 94.90 (0.12) | 25.97 (0.03) | 34.47 (0.11)   | 89.73 (0.22) |

Table 14: Comparisons on Image to Text Matching datasets. Mean and (Vairiance) are reported.

Flower102-text. Similar to CUB200, we use this dataset for Image to Image Matching, Text to Image Matching and Image to Text Matching tasks. For Image to Image Matching, we randomly select a pair of images and label them as matching pairs if they are from the same category. For Text to Image Matching and Image to Text Matching, we use the image-text pairs from Reed et al. [2016]

**MSCOCO** [Lin et al., 2014] is a large-scale dataset for object detection, point detection and captioning. It contains 118K training images and 5K validation images. Each image has 5 text descriptions. We used the 5K validation images to build the Image to Text Matching dataset. The train/val/test ratio is roughly 6/1/3.

**Flickr30K** [Young et al., 2014] is a popular benchmark for sentence-based picture portrayal. The dataset is comprised of 31,783 images that capture people engaged in everyday activities and events. Each image has 5 descriptive captions. This dataset is commonly used as a standard benchmark for Image to Text and Text to Image Matching.

**IPC (Image Paragraph Captioning)** contains 19,561 images from the Visual Genome dataset [Krishna et al., 2017]. Each image contains one paragraph describing the image. The training/val/test sets contains 14,575/2,487/2,489 images.

## G.2 Baseline Methods

We compare AutoMM with Sentence Transformer [Reimers and Gurevych, 2019] on semantic matching tasks. Sentence Transformer could compute embeddings for sentences, paragraphs, and images. The models are based on transformer networks like BERT / RoBERTa / XLM-RoBERTa / CLIP, etc. Text and image are embedded in vector space such that similar text and image are closer and can efficiently be found using cosine similarity. It also allows finetuning these embedding models on the target datasets to achieve maximal performance. For Text to Text Matching, we use its best text embedding model 'all-mpnet-base-v2'. For Image to Image, Image to Text and Text to Image Matching tasks, we use its best image-text embedding model 'CLIP-ViT-L-14'. We use its default hyperparameters for training. We run all the semantic matching experiments on a single NVIDIA A100 40G GPU.

## G.3 Metrics

For Text to Text and Image to Image Matching tasks, we use Area Under the Receiver Operating Characteristic Curve (ROC_AUC) as the evaluation metric. For Image to Text and Text to Image Matching tasks, we follow the literature [Sarafianos et al., 2019] to use Recall@K (R@K), which is defined as the portion of queries whose ground truth is within the top-K responses. Specifically, we use the average of R@K, where K=1, 5, 10, as the evaluation metric.

| Problem Type | #params (M) | #Trainable Params (M) | Peak Memory (MB) | Data Preprocessing Throughput (samples/s) | Training Throughput (samples/s) |
|---|---|---|---|---|---|
| classification/regression (Image + Text + Tabular) | 183 | 183 | 6032 | 111240.6 | 142.7 |
| semantic matching (TTM) | 109 | 109 | 8346 | 150947.2 | 362.3 |
| semantic matching (IIM) | 194 | 194 | 9460 | 110602.5 | 75.8 |
| semantic matching (ITM) | 427 | 427 | 20042 | 440922.1 | 489.3 |
| object detection | 218 | 218 | 22398 | 3480.7 | 14.1 |
| semantic segmentation | 645 | 9 | 25916 | 16224.8 | 22.2 |

Table 15: Computational Complexity Analysis of AutoMM (Part 1)

### G.4 Raw Results

We provide the detailed mean and variance of the model performance in Table 11-14. We repeat each experiment three times.

## H Computational Complexity Analysis of AutoMM

To analyze the computational complexity and performance of AutoMM, we present two tables (Table 15 and Table 16) that showcase various computational metrics for each problem type in an AWS EC2 P4d.24xlarge instance with 8x A100 40G GPUs. The datasets selected for this analysis are Memotion Mishra et al. [2023], MRPC [Dolan and Brockett, 2005], Airbnb [Airbnb, 2020], CUB200 [Wah et al., 2022], Comic [Inoue et al., 2018], and Leaf [Rath, 2023], with per gpu batch size of 1, 8, 8, 8, 1, and 1 respectively for problem types classification/regression, semantic matching (TTM), semantic matching (IIM), semantic matching (ITM), object detection, and semantic segmentation.

The number of parameters and peak memory usage vary significantly across problem types, indicating the diverse computational requirements of AutoMM. Data preprocessing throughput is generally high, ranging from 3480.7 to 440922.1 samples/s, demonstrating AutoMM's efficiency in handling data preprocessing tasks.

Training and validation throughput values provide insights into the model's efficiency during the learning process, with the highest training throughput observed for semantic matching (ITM) at 489.3 samples/s and the lowest for object detection at 14.1 samples/s. Inference throughput showcases the model's performance during the inference stage, with semantic matching (ITM) having the highest throughput at 9592 samples/s, and semantic segmentation having the lowest at 44.3 samples/s.

Result postprocessing throughput is generally high, ranging from 6523.2 to 183291085 samples/s, indicating AutoMM's efficiency in processing and outputting results across various problem types.

The computational complexity analysis demonstrates the efficiency and scalability of AutoMM across different problem types, highlighting its potential for real-world applications. The tables aim to offer insights into the computational requirements and performance of AutoMM, enabling informed decision-making when applying this approach to specific use cases.

| Problem Type | Validation Throughput (samples/s) | Inference Throughput (samples/s) | Inference FPS (single GPU single batch size) | Result Post-processing Throughput (samples/s) |
|---|---|---|---|---|
| classification/regression (Image + Text + Tabular) | 303.9 | 1165.3 | 35.6 | 183291085 |
| semantic matching (TTM) | 1716.3 | 4072.5 | 41.3 | 229010.1 |
| semantic matching (IIM) | 164.2 | 131.7 | 16.0 | 79449796.9 |
| semantic matching (ITM) | 1998.3 | 9592 | 41.5 | 102204.8 |
| object detection | 66.7 | 62.5 | 8.9 | 6523.2 |
| semantic segmentation | 57 | 44.3 | 6.9 | 15640442 |

Table 16: Computational Complexity Analysis of AutoMM (Part 2)

