# OpenReview forum: "AutoGluon-Multimodal (AutoMM): Supercharging Multimodal AutoML with Foundation Models"
_automl.cc/AutoML/2024/ABCD_Track — AutoML 2024 (ABCD Track)_

### Official Review · Reviewer_bgXh · 2024-03-22

**Potential Impact On The Field Of Automl Rating:** 2
**Technical Quality And Correctness Rating:** 3
**Clarity Rating:** 3
**Actions Required To Increase Overall Recommendation:** see above

**Summary Of Contributions:**

AutoGluon-Multimodal (AutoMM) is introduced as an open-source AutoML library designed for multimodal learning with foundation models. It claims to simplify the fine-tuning process and supports various data modalities and tasks. The paper presents experimental results to demonstrate AutoMM's superior performance in basic tasks and competitive results in advanced tasks compared to existing AutoML tools.

**Clarity:**

This paper is generally clear in its contributions and methodology. However, it could benefit from more detailed explanations in certain areas, such as the enhancement of interpretability and the practical applications of AutoMM.

**Overall Review:**

Paper Strength
1. Innovative Approach: The introduction of AutoMM for automating the fine-tuning of foundation models on multimodal data is commendable.

2. Broad Support: The library's support for diverse data modalities and tasks is a significant strength.

3. Performance: This paper presents strong experimental results showing that AutoMM outperforms existing AutoML tools in certain tasks.


Paper Weakness
1. Lack of Depth in Computational Complexity Analysis: This paper does not provide a thorough analysis of the computational complexity of AutoMM, which is essential for evaluating its efficiency in real-world scenarios.

2. Insufficient Discussion on Interpretability: This paper mentions interpretability as a strength but fails to provide a detailed comparison with existing methods, leaving the reader uncertain about the extent of this improvement.

3. Limited Real-World Applications: This paper lacks a demonstration of AutoMM's practicality through real-world applications, which would have strengthened its claims.

**Potential Impact On The Field Of Automl:**

While AutoMM has the potential to impact the field of AutoML positively by making multimodal data processing more accessible, the paper's limitations in providing a comprehensive analysis and real-world applications may hinder its adoption.

**Review Confidence:**

4

**Review Rating:**

6

**Review Summary:**

This paper presents a potentially valuable contribution to the AutoML field with AutoMM. However, to strengthen the paper, the authors should address the critical weaknesses, particularly by providing a more in-depth analysis of computational complexity and a clearer demonstration of interpretability improvements. Showcasing real-world applications would also enhance the paper's credibility.

----------------------
Some concerns are addressed. I have increased the rating.

**Technical Quality And Correctness:**

The technical approach seems sound, and the experimental results are promising. However, the paper's lack of depth in computational complexity analysis and interpretability discussion raises concerns about the thoroughness of the evaluation.

---

### Official Review · Reviewer_k4Qd · 2024-03-28

**Potential Impact On The Field Of Automl Rating:** 4
**Technical Quality And Correctness Rating:** 4
**Clarity Rating:** 3

**Summary Of Contributions:**

The paper presents an extension to the AutoGluon framework by adding support for multiple modalities.
The paper explains the use of the framework titled AutoMM.
The framework also includes tasks beyond classification and regression.
Multiple modalities are significant and a framework of this kind for Automated Machine Leaning for multiple modalities will be helpful.

**Actions Required To Increase Overall Recommendation:**

Inclusion of an exhaustive list of demos as part of supplementary material would have better and increase the readability and interest.

**Clarity:**

The documentation of the framework is clear. The description of APIs and commands for accessing the database are clearly mentioned.

**Overall Review:**

The paper presents a new framework for AutoML with multimodal datasets and includes models for tasks more complicated than classification and regression. It is integrated as part of an existing and well-maintained framework. It also hosts a vast collection of models as part of its model zoo giving the opportunity for the users to conduct exhaustive experiments. The document is clearly written and is apt for ABCD track.

**Potential Impact On The Field Of Automl:**

There will be significant impact of the proposed repository on the field of AutoML. The need for frameworks that are well maintained and up to date in the AutoML domain are necessary for progressive research.

**Review Confidence:**

5

**Review Rating:**

8

**Review Summary:**

I recommend that the paper be accepted for ABCD track. The paper describes the framework for automated machine learning with multiple modalities. They work does not include modalities of audio and video, however, there is only one another repository describing a similar work. A unique contribution of the proposed work is the inclusion of complex tasks like detection and segmentation rather than focusing only the well-searched and explored tasks like classification and regression. This enhances the scope of AutoML in these domains. Additionally, ML models with multiple modalities is the need of the hour. A framework for AutoML in this domain will only help the community in testing the algorithms that they are developing to cater to the growing trend of multi-modality. The repository will therefore be a useful contribution to the researchers in the domain of AutoML.

**Technical Quality And Correctness:**

The proposed benchmark is of high quality. It is proposed as an extension to an existing framework by including two primary features (1. multi modal benchmarks and 2. more complex tasks in Computer Vision) and encompassing the existing features of AutoGluon framework.

---

### Official Review · Reviewer_ikKS · 2024-04-02

**Potential Impact On The Field Of Automl Rating:** 3
**Technical Quality And Correctness Rating:** 3
**Clarity Rating:** 3
**Actions Required To Increase Overall Recommendation:** Addressing the bulletpoints in the "c…

**Summary Of Contributions:**

The paper introduces AutoMM, a library for multimodal learning with foundation models. At it's core AutoMM selects appropriate foundation models from existing model zoos, fine-tunes them to the task at hand, and uses a late-fusion model (in the case of multi-modal inputs) to create the final prediction. While selection mechanisms are used to select appropriate foundation models, hyperparameters are not tuned but remained fixed, based on a pre-study.

**Clarity:**

* More details are needed on how the selection mechanism works as this is a core element for AutoMM. E.g., are there "default" models that get picked form the zoos based on modality or is there a more refined selection mechanism behind AutoMM?
* If possible, more clarity needs to be given about the presets, how they were determined and how they differ (the appendix does not add additional explanations)
* Section 4.1 claims statistical significance, without ever stating which test was used.

**Overall Review:**

AutoMM seems like an interesting tool that could garner wider interest in the community. The work needs a bit of additional discussion of related work to better position itself and improvements in clarity as crucial details seem to be missing from the main body of the work.

**Potential Impact On The Field Of Automl:**

I believe the work could inspire many other multi-modal approaches and tools in the AutoML community.

**Review Confidence:**

4

**Review Rating:**

8

**Review Summary:**

Good paper with minor flaws.

**Technical Quality And Correctness:**

I believe the work would benefit from discussing other AutoML approaches aiming to utilize pre-trained models or foundation models. While the listed related work is relevant, discussing AutoMM in the context of foundation model use in and with AutoML would strengthen the paper. For example, [Quick-Tune](https://arxiv.org/abs/2306.03828) proposes how and which models to fine-tune from a pre-trained zoo. [TabPFN](https://openreview.net/forum?id=cp5PvcI6w8_) proposes to use transformers to solve tabular classification problems or [OptFormer](https://arxiv.org/abs/2205.13320) that proposes to learn universal hyperparameter optimizers. I believe including such works for an extended discussion of how foundation models are transforming the AutoML landscape (see, e.g. [Tornede et al. (2024)](https://openreview.net/forum?id=cAthubStyG)), would strengthen the paper as it would even closer highlight the lack of multi-modal approaches by the community.

In the Section 3.4 the following is stated:
> By default, AutoMM trains only one late-fusion model with fixed hyperparameters (without optimization). These default hyperparameters, e.g., learning rate and weight decay, have been pre-determined through offline search across diverse benchmark datasets, ensuring they are likely to yield reasonable performance on new (similar) datasets. This approach differs from many previous AutoML toolboxes [Wang et al., 2021a, Erickson et al., 2020], which typically emphasize either hyperparameter optimization or model ensembling/bagging.

I am not sure I understand this correctly. Is this offline tuning approach similar to Autosklearn 2.0, in which reasonable "default" hyperparameters are determined on a large variety of datasets (however Autosklearn 2.0 uses those then as starting points for further HPO)? In either case, I believe this statement needs to be clarified a bit as AutoML tools have often employed reasonable default hyperparameters from which to start further HPO.

---

### Meta-Review · Area_Chair_YT9Z · 2024-04-23

**Paper Recommendation:** Accept
**Confidence:** 4

**Metareview:**

The paper proposes a library for multimodal learning with foundation models. All reviewers support acceptance. After carefully reading all reviewers' comments, I recommend acceptance and strongly suggest the authors address the reviewers' comments in the final paper version.

---

### Decision · Program_Chairs · 2024-04-29

**Decision:**

Accept

**Comment:**

Thank you for submitting your paper. We are happy to tell you that we accept your paper to the main track. See you in Paris.